# Research on defect detection of bottle cap interior based on low-angle and large divergence angle vision system

**Bowen Chen, Chen Li\*, Pi Yuan, Yujie Yan, Yongjing Yin**

School of Mechanical and Energy Engineering, Zhejiang University of Science and Technology, Hangzhou, China

\* lee_lichen@163.com

## Abstract

During the machine vision inspection of the inner section of bottle caps within pharmaceutical packaging, the unique conca bottom and convex side walls often create obstructions to the illumination. Consequently, this results in challenges such as irregular background and diminished feature contrast in the image, ultimately leading to the misidentification of defects. As a solution, a vision system characterized by a Low-Angle and Large Divergence Angle (LALDA) is presented in this paper. Using the large divergence angle of LED, combined with low-angle illumination, a uniform image of the side wall region with bright-field characteristics and a uniform image of inner circle region at the bottom with dark-field characteristics are obtained, thus solving the problems of light being obscured and brightness overexposure of the background. Based on the imaging characteristics of LALDA, a multi-channel segmentation (MCS) algorithm is designed. The HSV color space has been transformed, and the image is automatically segmented into multiple sub-regions by mutual calculation of different channels. Further, image homogenization and enhancement are used to eliminate fluctuations in the background and to enhance the contrast of defects. In addition, a variety of defect extraction methods are designed based on the imaging characteristics of different sub-regions, which can avoid the problem of over-segmentation in detection. In this paper, the LALDA is applied to the defect detection inside the cap of capsule medicine bottle, the detection speed is better than 400 pcs/min and the detection accuracy is better than 95%, which can meet the actual production line capacity and detection requirements.

## 1. Introduction

Pharmaceutical packaging safety [1,2] is closely related to the life and health of patients. The entry of foreign substances contaminating pharmaceutical into the human body can lead to adverse reactions [3–6]. For example, the black stains inside the cap of a capsule medicine bottle can contaminate the composition of the medicine, so it is necessary to detect defects in this pharmaceutical packaging [7–11]. In actual production, bottle caps with defects mainly include the mixed color within the same batch, black spots on the sidewall, as well as wrinkles,

**Data Availability Statement:** Website: https://figshare.com/account/home#/data Account: fishviews@163.com Password: Zxc.159357 This article involves a lot of product data within the

factory, so we will partially disclose the original data involved in the paper.

**Funding:** Funding provided by National Natural Science Foundation of China, 62103340 to CL.

**Competing interests:** The authors have declared that no competing interests exist.

missing, black spots, hair and scratches on the bottom caps [12–14], as shown in Fig 1. Both region A and Region B are internal regions of the cap to be detected. Region A is the side wall of the cap, and region B is the bottom inner circle of the cap (containing the smooth gasket).

In industry, defect detection typically utilizes technologies such as industrial cameras and 3D stereo cameras [15,16]. However, due to the complexity and higher cost of 3D stereo cameras, bottle cap detection commonly employs industrial cameras combined with high-angle ring lights or coaxial lights for inspection, along with different algorithms, to detect the foreign substances mentioned above [17,18]. However, the existing technologies have the following problems:

On the one hand, in traditional means for internal detection of bottle caps, due to the lower height of the bottom inner circle region compared to the side wall, as shown in Fig 2(a), it is easy to create light path obstruction during imaging. This causes the middle region to converge more light and the surrounding area less light. This produces the problem of higher brightness in the middle region and lower brightness in the surrounding (as shown in (b) and (c)), and even reduces the contrast of the image, enhancing the difficulty of defect recognition. Further, in order to enhance the image brightness of the side wall region of the bottle cap, the illumination of the incident light source can be enhanced. But it will cause the overexposure of the brightness of the inner circle region, which tends to mask the defect, as shown in (d). In addition, in order to solve problem such as uneven image, some studies have drilled holes in the conveyor belt and installed a planar light source below the belt [19]. At the same time, utilizing a ring light above the cap with the auxiliary lighting at the bottom, can better reveal the defects in the bottle cap. But when it comes to thicker caps, this kind of auxiliary lighting cannot play a very good advantage.

On the other hand, various detection algorithms, such as deep learning and traditional algorithms, are often used in the later stage of image recognition [20–25]. However, due to the wide variety of colors, shapes, and sizes of bottle caps, achieving comprehensive detection using traditional recognition algorithms is challenging. Therefore, the more commonly used method is the defect detection method based on machine learning [22,25–29]. However, deep learning requires a large number of samples with defects for training, and necessitates the collection of new samples for each type of new bottle cap. Obtaining a sufficient number of samples within a short timeframe during real production is difficult, leading to lower recognition accuracy [30,31]. That is, the methods of deep learning can further improve the generality of the detection system, but at the beginning of the project, due to the high production yield in the actual scenario, there are fewer samples with defects, which will result in lower detection accuracy. Therefore, the design of the defect recognition algorithm in this paper will be completed for the characteristics of the defect, based on the classical algorithm and according to the obvious feature differences. In addition, traditional algorithms are also widely used in

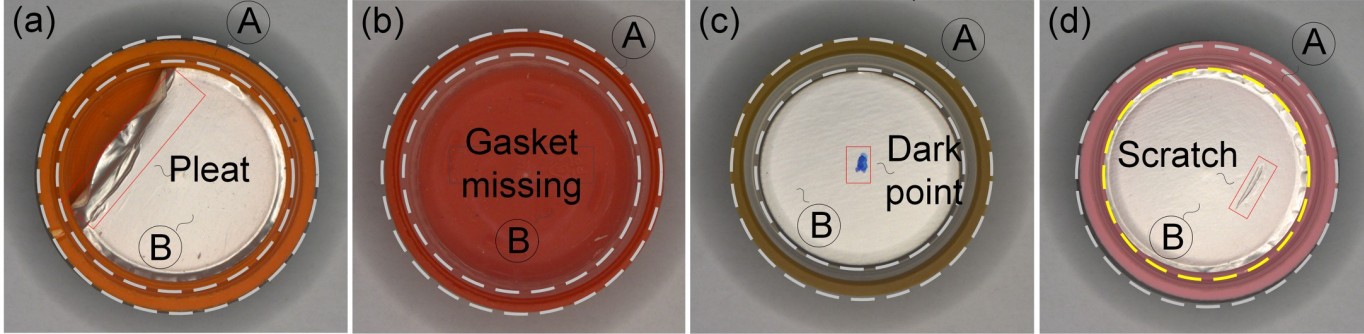

**Fig 1. Types of defects to be detected.** (a) gasket pleat; (b) gasket missing;(c) dark point;(d) scratch.

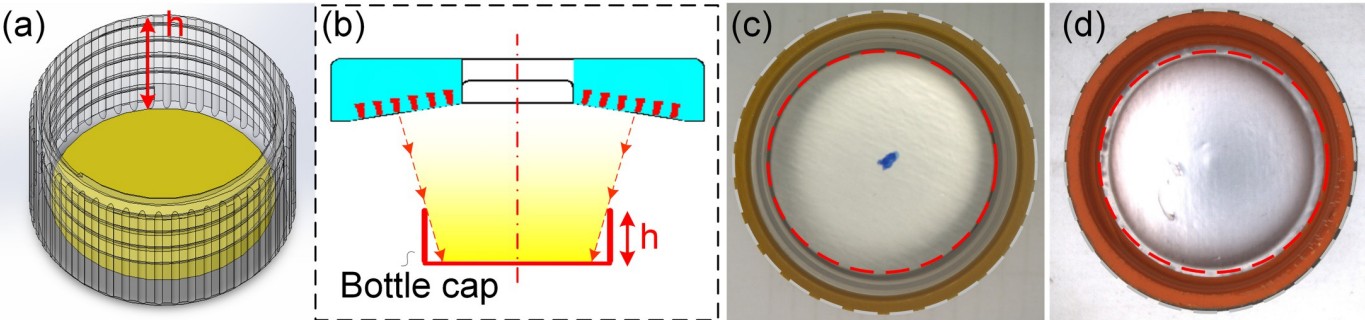

**Fig 2. The structure of the bottle cap to be detected.** (a) 3D schematic of the bottle cap; (b) commonly used high-angle ring light detection method; (c) the image brightness of the side wall region of the cap is dark, and the image of the inner circle region also has the problem of uneven background; (d) the image brightness of the side wall region is dark, but the brightness of the inner circle region is overexposed.

bottle cap detection [32], however, the existing traditional algorithms need to do a lot of calculations for the background and require much manual time to draw the detection region and manually divide the image into multiple sub-regions. Different detection methods are then used to process different sub-regions separately, thus can avoid over-segmentation or misclassification of defects in the global image processing.

Based on the above analysis, for the detection of the inside of the medicine bottle cap, on the one hand, there is a lack of a uniform imaging detection scheme, to be able to illuminate the bottle cap uniformly and obtain a uniform image. On the other hand, there is a lack of a more general detection algorithm that can efficiently segment the defects. Therefore, this paper proposes a vision inspection system with low angle and large divergence angle (LALDA) for the detection of defects inside the bottle cap. Using the large divergence angle of the LED, combined with low-angle illumination, it is possible to obtain uniform bright-field imaging of the side wall and uniform dark-field imaging of the bottom inner circle region. This can overcome the shortcoming of the traditional inspection schemes in which the side wall of the cap and the inner circle of the bottom cannot be uniformly illuminated at the same time. Then, a multi-channel segmentation (MCS) algorithm is designed to automatically segment the interior of the bottle cap into multiple sub-regions based on the imaging characteristics of different regions inside the bottle cap, combined with the mutual calculation between different image channels in HSV color space. The image homogenization and enhancement operators are further designed to enhance the defect information and eliminate the complex background as well as reduce the complexity of the algorithm, which can avoid problems of over-segmentation and misclassification in defect detection. The innovative application of this paper consists of the following main aspects:

1. A machine vision inspection system with low angle and large divergence angle has been built, to avoid the obstruction of the illumination of the bottom inner circle of the bottle cap by the side wall with high height, and to solve the problem that different areas inside the bottle cap cannot be illuminated uniformly at the same time. That is, the side wall region of the cap achieves uniform bright-field illumination, and the bottom inner circle region achieves uniform dark-field illumination. The system can avoid large brightness fluctuations in the image background, as well as the problems of too dark gray values and overexposure of brightness in the background.

2. Based on the above detection system, a multi-channel segmentation (MCS) algorithm is designed. In the HSV color space, different channels are mutual calculated to achieve adaptive segmentation of sub-regions to be detected, thus avoiding the traditional algorithm

that requires a lot of mutually operations such as complex region selection. At the same time, the background homogenization and enhancement operations can reduce the brightness fluctuation of the background as well as enhance the contrast of defects. Further, different recognition algorithms are designed according to the imaging characteristics of different sub-regions. And the feature extraction of defects can be accomplished by adaptive global threshold segmentation in different sub-regions. Thus, it avoids the problem of deep learning, which requires the collection of a large number of samples with defects, and its poor applicability to the detection of new defects or new samples.

The remainder of this paper is organized as follows. Section 2 mainly focuses on the construction of the LALDA system and introduces the basic components of the system. Compared with the traditional detection scheme, the LALDA can obtain a more uniform image. Section 3 focuses on the algorithm study of MCS, which converts RGB image into HSV image, and uses different channels of HSV to segment different sub-regions. The image homogenization, enhancement and recognition methods are also designed for different sub-regions and different defects. In section 4, experimental verification is performed according to the different defects and the different types of caps. Finally, a brief conclusion is presented in Section 5.

## 2. Detection system based on low-angle and large divergence angle (LALDA)

This thesis focuses on the detection of defects inside pharmaceutical bottle caps. Traditional inspection solutions often use high-angle ring light or coaxial light to illuminate the bottom of the bottle cap. However, because the cap has the characteristics of low height in the middle and high height on the side, it will produce obscuration of the illumination of the inner circle region at the bottom, thus producing uneven image background. Furthermore, the incident light fails to directly illuminate the side wall, leading to reduced brightness in the corresponding image area. This dimness increases the likelihood of oversight in defect detection. It also causes the problem of overexposure of the brightness or low grayscale value of the sub-regions to be detected. In view of the above problems, this paper adopts a low-angle illumination method by raising the height of the light source beyond the theoretical working distance. It further controls the brightness intensity, allowing the incident light to directly and uniformly illuminate the side region of the bottle cap under inspection. This approach overcomes the problem of low brightness of the image on the side wall. In addition, using the large divergence angle of the LED, it can illuminate the inner circle region of the cap evenly and form a dark-field illumination when illuminating the side region. Further, the working distance of the lens is reduced and a lens with a small focal length is used to achieve simultaneous imaging of the sub-regions, and a large field angle is used to increase the imaging information on the side. Based on the above analysis, this paper designs a vision inspection system with low-angle and large divergence angle (LALDA), as shown in Fig 3.

In the inspection system shown in Fig 3, the height and intensity of the light source are adjusted so that the low-angle light source is directed to the side of the bottle cap. Excessive or insufficient brightness of the light source can lead to overexposure or underexposure of the side and bottom of the bottle cap, resulting in indistinct defect features. This makes it difficult to segment the bottle cap area and extract defect features during later algorithm processing. The light source selected in this paper with the incidence angle of about $15° \sim 30°$. In the conventional use of the light source, the height of the lower surface of the light source from the surface to be detected is low enough to ensure uniform illumination of the plane to be detected (such as uniformly illuminated plane P in Fig 3). However, in this paper, the height of the low-angle light source is raised, and the incident light is directly incident on the side wall of the bottle cap, so as to obtain a uniform image of the side wall region. In addition, due to the existence

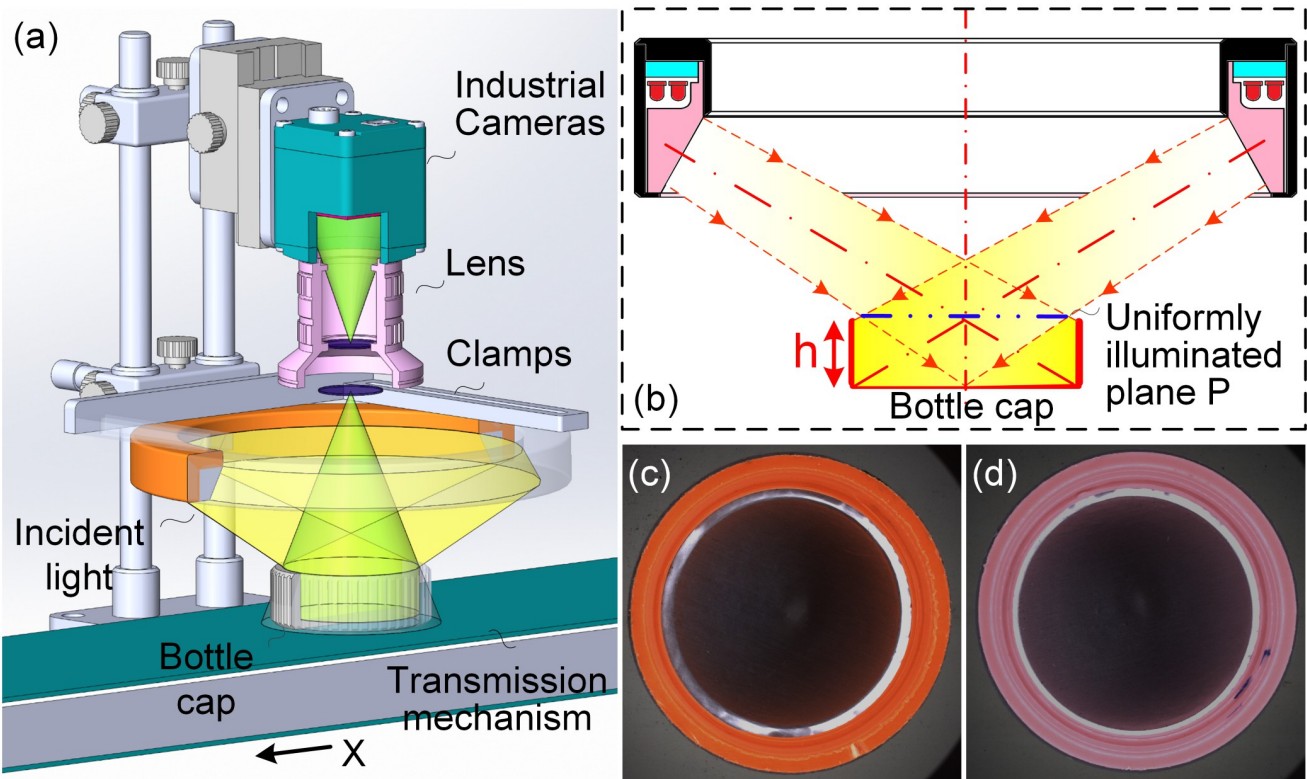

**Fig 3. The detection system of LALDA.** (a) system schematic; (b) basic principles of low-angle incident light illumination; (c)(d) images of bottle caps in different colors captured by the system.

of the gasket with smooth surface in the inner circle of the bottle cap, after the incident light at a low angle is reflected from the surface to be inspected, the reflected light cannot enter the imaging system. Finally, an imaging lens with a small focal length is utilized, with a large field angle, thus getting more information of the side wall. And the whole system can get the diffuse reflected light from the side wall of the cap, as well as the scattered light from the defects in the inner circle region of the bottom.

In summary, by the above light condition, the shading of the inner circle of the bottom by the side wall of the cap is avoided. And it can avoid the image brightness of the side wall is too dark, while the image of the inner circle has an uneven brightness, or even brightness overexposure. Using the above system, on the one hand, for the side wall region of the cap, a uniform bright-field image is formed; on the other hand, for the inner circle region of the cap, a uniform dark-field image with black background and white defects is formed. Therefore, in the following, the algorithms will be designed separately for the imaging characteristics of different sub-regions to be detected, and for different defects.

## 3. Defect detection algorithm based on multi-channel segmentation (MCS)

The LALDA vision system can avoid incident light being blocked by the side of the bottle cap, avoiding overexposure or too dark background in the image. That is, there are two regions of uniform imaging inside the cap, the side wall region with bright-field characteristics, and the inner circle region with dark-field characteristics. This paper draws on the traditional

detection algorithm to divide the image to be processed into two sub-regions and further design different defect extraction algorithms. However, the traditional algorithms use manual drawing of different sub-regions. On the one hand, it takes more time, and on the other hand, due to the motion error of the conveyor belt, the region to be detected in the image will change when detecting the caps one by one. Thus, the regions to be detected and the fixed drawn regions will have position deviation, which affects the accuracy of recognition. To address this problem, this paper designs an adaptive region segmentation algorithm, namely the multi-channel segmentation (MCS), which avoids complex manual region drawing and adaptively segments sub-images, ensuring that the sub-regions can also be accurately segmented when the position of the product fluctuates. Then different detection algorithms are designed according to the imaging characteristics of different sub-regions, different background fluctuations, and different defects to be detected. The entire algorithm flow is shown in Fig 4.

The interior of the bottle cap to be detected contains a side wall region with bright-field characteristics, and an inner circle region with dark-field characteristics. Based on the above analysis, the above two sub-regions are first partitioned.

Since the color space of the image acquired by the system is RGB, the traditional methods will first convert the RGB image to grayscale image, and then complete the segmentation of different sub-regions based on the grayscale image. One of the commonly used grayscale

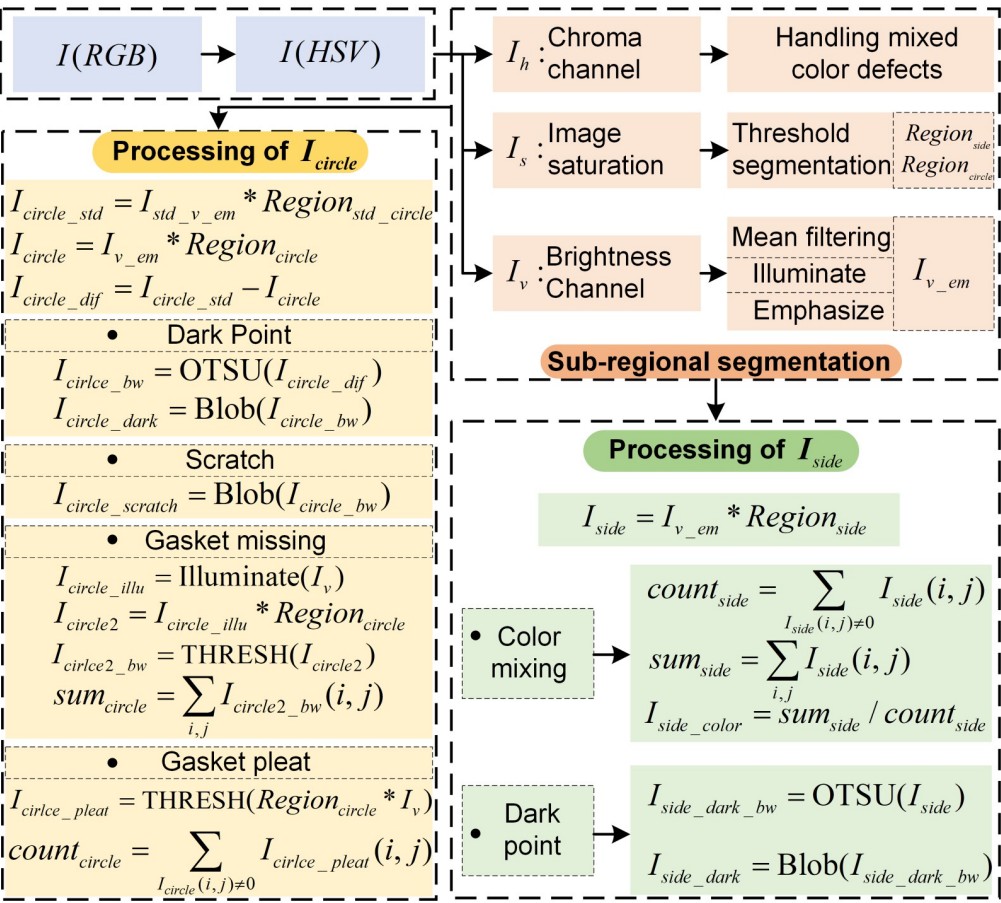

**Fig 4. Algorithm flow.**

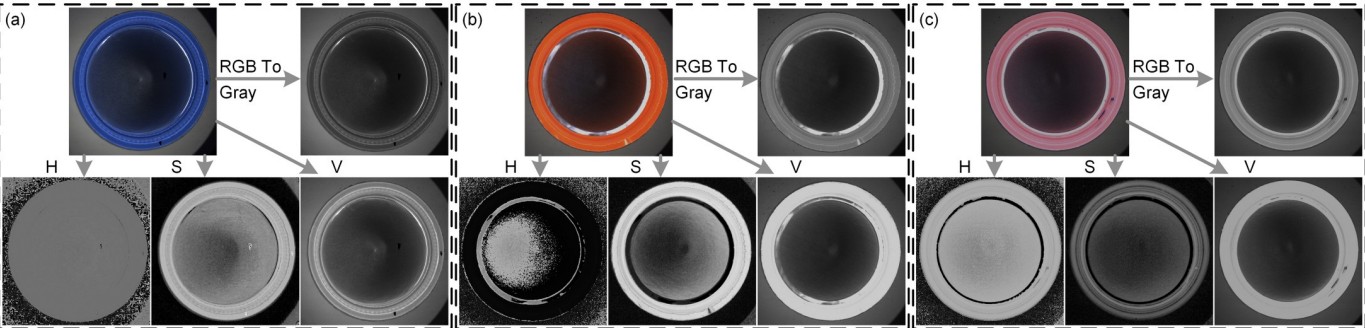

**Fig 5.** Direct grayscale operation of different objects to be detected, as well as conversion to HSV color space: (a) the conversion of blue bottle cap; (b) the conversion of red bottle cap; (c) the conversion of pink bottle cap.

operations is shown in Eq (1):

$$I_{gray} = I_r * 0.299 + I_g * 0.587 + I_b * 0.114 \tag{1}$$

However, since some caps have the property of color bias, i.e., only one of the three channels, R, G and B, has values, the brightness will be averaged out by the above calculation. That is, the luminance value is lower in the grayscale image, thus reducing the contrast of the image (as in the RGB_To_GRAY operation in Fig 5). To address this problem, this paper first uses HSV-based color space based on the chromaticity, luminance, and other features to divide different sub-regions. Since the H-channel (the hue of the image) retains the color information of the original image, it can be used to determine whether there is color mixing defect. In addition, since the S-channel retains the saturation characteristics of the original image, it is possible to segment the internal region of the bottle cap with brightness fluctuations. Further, since the V-channel (luminance channel), which is rich in detail information, the identification of defects can be completed in the V-channel. The specific process is shown below.

**(1) Sub-regions segmentation based on HSV**

Based on the above analysis, the RGB image is first converted to HSV image:

$$I_h = \begin{cases} 0^\circ & if\ I_{max} = I_{min} \\ 60^\circ \times \dfrac{I_g - I_b}{I_{max} - I_{min}} + 0^\circ & if\ I_{max} = I_r \& I_g \geq I_b \\ 60^\circ \times \dfrac{I_g - I_b}{I_{max} - I_{min}} + 360^\circ & if\ I_{max} = I_r \& I_g < I_b \\ 60^\circ \times \dfrac{I_b - I_r}{I_{max} - I_{min}} + 120^\circ & if\ I_{max} = I_g \\ 60^\circ \times \dfrac{I_r - I_g}{I_{max} - I_{min}} + 240^\circ & if\ I_{max} = I_b \end{cases} \tag{2}$$

$$I_s = \begin{cases} 0 & if\ I_{max} = 0 \\ \dfrac{I_{max} - I_{min}}{I_{max}} = 1 - \dfrac{I_{min}}{I_{max}} & otherwise \end{cases}$$

$$I_v = I_{max}$$

Where, $I_h$ is the hue. $I_s$ is the saturation. $I_v$ is the brightness. $I_r$, $I_g$, $I_b$ are the three channels of the image. The maximum and minimum values of the grayscale are $Imin_{max}$. The above conversion process is shown in Fig 5.

As can be seen in Fig 5, the V-channel has higher contrast and more detail than the image directly operated by graying (as Eq (1)), and preserves the bright-field and dark-field features in the original image compared to the other H and S channels. It should be noted that in the V-channel in (a), the difference between the gray value of the whole cap region and the external region of the cap is small, and the region to be detected cannot be separated directly from the V-channel. However, it can be seen from the S-channel that the whole cap region has obvious contrast with the surrounding background, which can be quickly segmented by a simple grayscale threshold segmentation. In addition, since the V-channel retains the bright grayscale features of the side wall region and the dark grayscale features of the background of the inner circle region, the inner circle region ($Region_{circle}$) can be quickly segmented using adaptive binarization threshold segmentation and simple image processing. Then the side wall region ($Region_{side}$) is quickly calculated by combining the entire cap region segmented by the above S-channel.

Further, pre-processing operations of the image need to be performed on the V-channel before the defect extraction. That is, the noise can be removed using mean filtering, and then the background homogenization is performed by Eq (3) to equalize the brightness of the image.

$$I_{v\_illu} = \text{round}((127 - mean_{m \times m}) * K_{illu}) + I_v \tag{3}$$

Where, $I_{v\_illu}$ is the image after homogenization, $round$ is the rounding function, $mean_{m \times m}$ is the mean value within the window size $m \times m$, and $K_{illu}$ is the enhancement factor (taken as $K_{illu} < 1$).

By the above operation, the filtered image is compared with the median gray value 127 of the image, and the difference is multiplied by $K_{illu}$. It is able to make the local grayscale close to the median gray value to achieve the effect of homogenizing the image background.

In order to enhance the contrast of the defects in a uniform background, Eq (4) is used for contrast enhancement.

$$I_{v\_em} = \text{round}((I_{v\_illu} - mean_{m \times m}) * K_{em}) + I_{v\_illu} \tag{4}$$

Where, $I_{v\_em}$ is the enhanced image, $mean_{m \times m}$ is the mean value in the image $I_{v\_illu}$ within the window size $m \times m$, and $K_{em}$ is the enhancement factor ($K_{em} > 1$).

Using the above operation, it is possible to compare the grayscale value of $I_{v\_illu}$ with the average value within the filter window, and magnify the difference, thus having the effect of enhancing the contrast.

Through the above preprocessing, in the V-channel, it can obtain an image with uniform background and strong contrast, and segment sub-regions ($I_{side}$ and $I_{circle\_std}$) according to $Region_{circle}$ and $Region_{side}$, as shown in Eq (5).

$$I_{side} = I_{v\_em} * Region_{side}, I_{circle} = I_{v\_em} * Region_{circle} \tag{5}$$

Subsequent image processing will be performed for each of the sub-regions, i.e., defect detection in the side wall region of the bottle cap and defect detection in the inner circle region of the bottom.

**(2) Detection of defects in the side wall region**

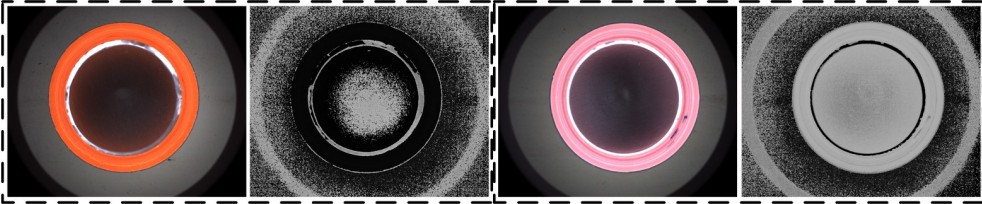

**Fig 6. Different colors and H-channel images after HSV.**

For the detection of defects in the side wall region, it is mainly for the detection of black points. In addition, in actual production there will be color mixing defect, i.e. caps of different colors will be mixed in the type of cap to be inspected.

(a) Detection of color mixing defect

The H-channel retains the color information of the image. In Fig 6, it can be seen from the H channel of different color bottle caps that the average values of hues can be very different. And the variability of color differences in the H-channel is much larger than the variance values in the grayscale image. Therefore, in this paper, the average gray value ($I_{side\_color}$) at the non-zero position of the gray level in the side wall region to be detected is calculated. And compared with $I_{side\_color}$ of the color type to be inspected to determine whether there is a color mixing defect.

First, the total number of non-zero pixels in the region is calculated according to Eq (6).

$$count_{side} = \sum_{I_{side}(i,j) \neq 0} I_{side}(i,j) \tag{6}$$

The sum of the grayscale values of the $I_{side}$ in the H-channel is then calculated as in Eq (7).

$$sum_{side} = \sum_{i,j} I_{side}(i,j) \tag{7}$$

Calculate the average gray value of non-zero elements according to (6)(7):

$$I_{side\_color} = sum_{side}/count_{side} \tag{8}$$

By the difference of $I_{side\_color}$ between the bottle cap to be detected and the color type of cap to be detected, it can quickly distinguish whether there is a color mixing defect.

(b) Detection of dark point

It should be noted that, compared with the traditional detection scheme, the detection system of LALDA proposed in this paper makes the side wall region uniformly illuminated. The brightness of the image is more uniform and the contrast is higher, which is shown as Fig 7. In addition, the background of the image is further homogenized and the contrast of defects is further enhanced by Eqs (3) and (4). Thus, the difficulty of algorithm design is greatly reduced when detecting dark point with low brightness. According to the actual detection process, the extraction of the dark point can be completed by adaptive global threshold segmentation. Further feature analysis based on the Blob can achieve the identification of dark point with the

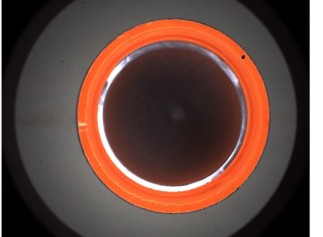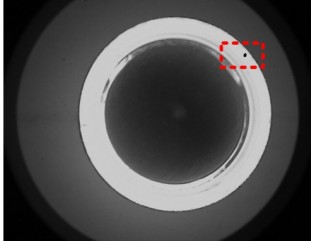

**Fig 7. Sample with dark point and the V-channel in the side wall region.**

process shown in (9)(10).

$$I_{side\_dark\_bw} = \text{OTSU}(I_{side}) \tag{9}$$

$$I_{side\_dark} = \text{Blob}(I_{side\_dark\_bw}) \tag{10}$$

Where, *OTSU* is the adaptive global threshold segmentation operation; the extraction and analysis of connected regions are achieved by *Blob* in the binarized image. The above two operations are not introduced in this paper.

Through the above process, the color mixing in the side wall region $I_{side}$ is detected by the average gray value of the non-zero elements ($I_{side\_color}$). And in the homogenized and enhanced image, operations such as adaptive global threshold segmentation can be used to achieve fast extraction of dark points in the region.

**(3) Detection of defects in the inner circle region**

For defect detection in the inner circle region at the bottom of the cap with smooth gasket, the gasket surface is inspected for dark point, gasket pleat, gasket missing, and scratch. The commonly used method is to make a template in a standard image without defects, and then the template is subtracted from the corresponding region in the image to be detected to extract the defects. However, in the process of acquiring standard image, due to the transmission error of the system, there will be deviations in the position of the area to be detected. And the standard images acquired at different moments will be different. This causes problems such as over-segmentation of the image when subtracting from the image to be detected. To avoid this problem, it is common practice to capture multiple images without defects, compute their average values, and utilize this average image as the reference standard. However, the calculation result of template is affected by the number of acquired images, and the process is less efficient. In addition, when there are brightness fluctuations in the image background, interference still cannot be avoided using the conventional method. Therefore, this paper addresses this problem by acquiring a single image without defects and producing a standard template image through operations such as background homogenization as well as image enhancement. Moreover, the same processing is applied to the image to be detected, and then the subtraction is done. In addition, because the background of the inner circle region has the characteristics of dark-field illumination, as well as different defects have different imaging characteristics, so it is necessary to design different detection algorithms for different targets to be detected.

(a) Inner circle template at the bottom of bottle cap

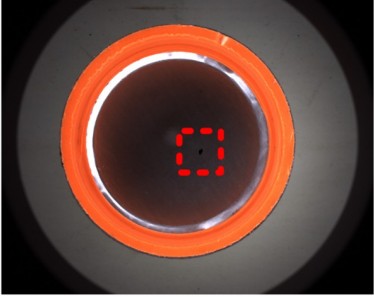
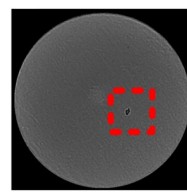

**Fig 8. Sample with dark point and the V-channel in the inner circle region.**

For the cap without defects, the mean filtering, background homogenization, and feature enhancement mentioned above are performed to obtain $I_{std\_v\_em}$. Then, inner circle template $I_{circle\_std}$ is calculated based on the $Region_{std\_circle}$.

$$I_{circle\_std} = I_{std\_v\_em} * Region_{std\_circle} \quad (11)$$

In addition, for the image to be detected, the inner circle image ($I_{cicle}$) is obtained using the same processing.

$$I_{circle} = I_{v\_em} * Region_{circle} \quad (12)$$

(b) Detection of dark point

Since the incident light is not parallel, its diverging light will enhance the brightness of the inner circle region. Thereby, the dark point has lower gray value or produce darker shadow compared to the gray background which is shown as Fig 8, so the difference in gray values can be calculated by the image subtraction.

$$I_{circle\_dif} = I_{circle\_std} - I_{circle} \quad (13)$$

Further, for the result of subtraction, adaptive grayscale segmentation and Blob analysis are used to enable fast extraction of defects ($I_{circle\_dark}$).

$$I_{circle\_bw} = \text{OTSU}(I_{circle\_dif}) \quad (14)$$

$$I_{circle\_dark} = \text{Blob}(I_{circle\_bw}) \quad (15)$$

(c) Detection of scratch on the gasket

Scratch appears as white in the image, and black features are formed at locations of outer contour with large changes in brightness after the image enhancement which is shown as Fig 9. Thereby, this black information is extracted from the result $I_{circle\_dif}$ and $I_{circle\_bw}$ after subtraction, and the scratch with linear characteristics is further extracted by Blob analysis.

$$I_{circle\_scratch} = \text{Blob}(I_{circle\_bw}) \quad (16)$$

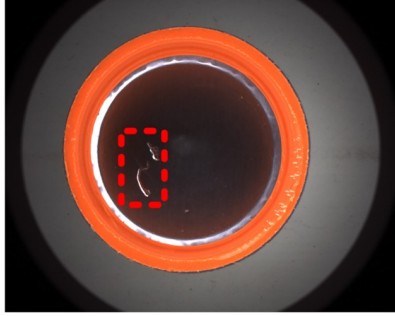
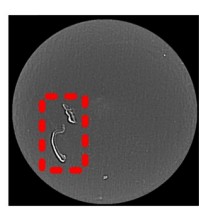

**Fig 9. Sample with scratch and the V-channel in the inner circle region.**

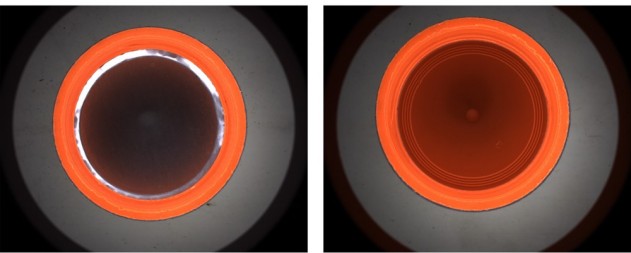

**Fig 10. Sample without defect and sample with gasket missing in the inner circle region.**

(d) Detection of gasket missing

For the bottle cap with gasket missing, the background has a large brightness fluctuation, i.e., the uniform background of the dark field is destroyed and a brighter background is formed which is shown as Fig 10. Therefore, it is possible to determine whether there is a defect of gasket missing based on the extent to which the dark area is destroyed. In this article, the Eq (3) is used to do the reverse operation. That is, $K_{illu}$ is set to a larger value, the grayscale values below the median value 127 are enhanced to a larger one which form bright spots. And the size of the bright spots in the inner circle region is calculated, which indicates the extent to which the dark field is destroyed. Then the defect of gasket missing is judged according to the size.

First, a reverse background correction is used to lighten the darker area of the image:

$$I_{circle\_illu} = \text{Illuminate}(I_v) \tag{17}$$

Where *Illuminate* is the operation of Eq (3), and $K_{illu} > 1$.

Then, the region to be processed $I_{circle2}$ has been taken out:

$$I_{circle2} = I_{circle\_illu} * Region_{circle} \tag{18}$$

Finally, a binarization is performed and the size of the area with bright background is calculated.

$$I_{cirlce2\_bw} = \text{THRESH}(I_{circle2}) \tag{19}$$

$$sum_{circle} = \sum_{i,j} I_{circle2\_bw}(i,j) \tag{20}$$

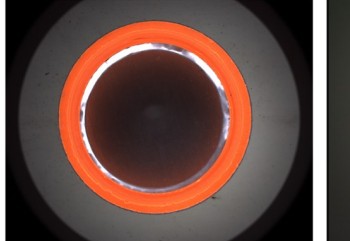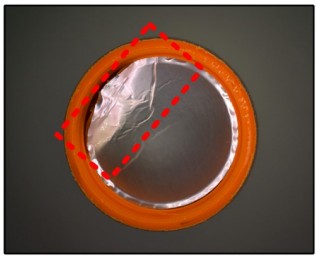

**Fig 11. Sample without defect and sample with gasket pleat in the inner circle region.**

By the above calculation, it can transform the dark background in dark-field imaging into bright, and obtain a large connected area by binarization. However, the gasket missing destroy the dark field imaging. And the connected area is smaller after the above calculation. The change in the size of the connected area enables a quick determination of the gasket missing.

(e) Detection of gasket pleat

Since gasket pleat produces texture changes as well as height variations that disrupt the dark-field imaging, it is easy to have areas of overexposure of brightness in the image which is shown as Fig 11. Therefore, in the inner circle region, the extraction of bright features is performed directly using threshold segmentation:

$$I_{cirlce\_pleat} = \text{THRESH}(Region_{circle} * I_v) \tag{21}$$

Further calculation of the size of the luminance overexposure area enables identification of large gasket pleat, as shown in Eq (22):

$$count_{circle} = \sum_{I_{circle}(i,j) \neq 0} I_{cirlce\_pleat}(i,j) \tag{22}$$

In summary, a multi-channel segmentation algorithm (MCS) is designed in this paper based on the imaging characteristics of the bottle cap, i.e., the side wall region has uniform bright-field characteristics and the bottom inner circle region has uniform dark-field characteristics. The original RGB color space is converted into HSV color space, and the adaptive segmentation of the sub-regions to be processed can be accomplished by using the characteristics of different channels and the mutual calculation between different channels. Then, the background in sub-regions is further homogenized and the contrast of defects is further enhanced by homogenization and enhancement operations. Thus, it is easy to extract the target to be detected quickly by global threshold segmentation directly in the subsequent calculation. Finally, according to the different kinds of defects have different imaging characteristics, different recognition algorithms have been designed respectively, and then fast and effective defect recognition can be achieved.

## 4. Experiments

Based on the analysis above, the detection system based on LALDA is first built. Using a low-angle light combining the large divergence angle of LED and a small focal length lens, uniform imaging of the inside of the bottle cap is achieved. And the bright-field imaging characteristics are formed in the side wall region of the cap, and the dark-field imaging characteristics are

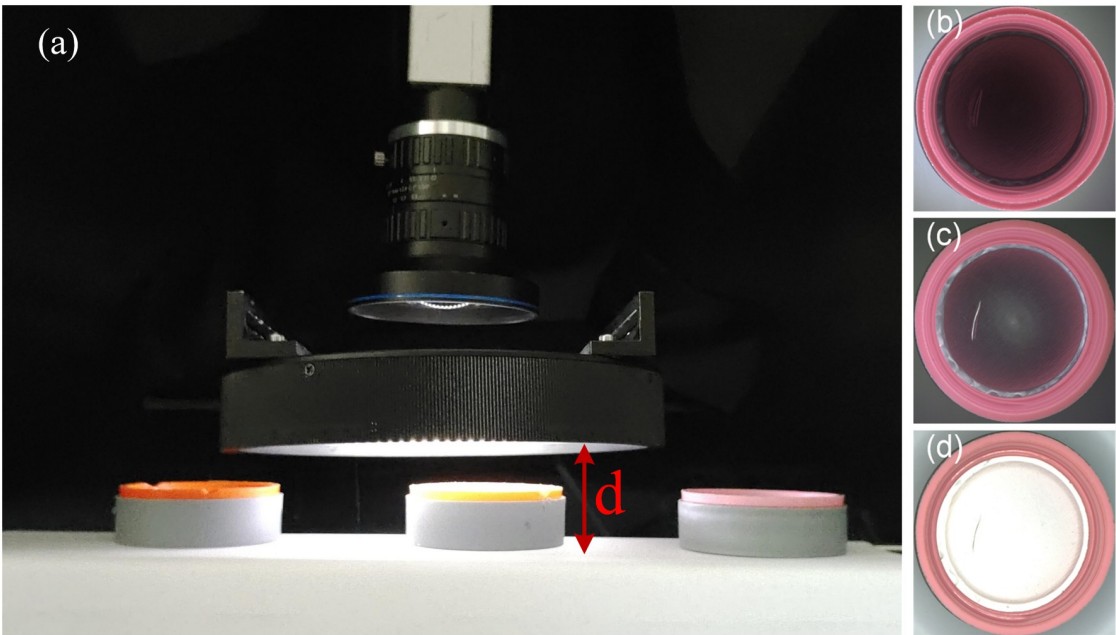

**Fig 12. The actual built detection system based on LALDA.** (a) the main components of the detection system; (b), (c), (d) imaging results of the light source at different working distances (d1, d2, d3).

formed in the bottom inner circle region. The inspection system shown in Fig 12. The actual built detection system based on LALDA: (a) the main components of the detection system; (b), (c), (d) imaging results of the light source at different working distances (d1, d2, d3). has an imaging unit with an 8mm lens and a 1.3 MP area scan camera. The illumination unit uses a low-angle light source of about 15°, and, when a uniform illumination is formed in the plane being inspected, its theoretical working distance (the distance between the lower surface of the light source and the detected plane) is about $d_1 = 20mm$.

When the light source is at the theoretical working distance, due to the small incident angle, the incident light is blocked by the side wall of the cap, thus unable to enter the bottom of the cap, resulting in internal defects that cannot be illuminated. According to the analysis in Section II, this paper aims to obtain a uniform background in the image, and gradually raises the height of the light source and until the side wall region of the bottle cap is uniformly illuminated, as well as the formation of a dark-field illumination in the bottom inner circle region. When the height of the incident light source is approximately $d_2 = 40mm$, the system creates a dark field illumination (black background and white defect) in the inner circular area of the cap and the incident light can directly illuminate the side wall area. If the height of the light source is further increased ($d_3 > 60mm$), most of the light is incident on the outside of the cap, and a small portion of the light is incident on the inside of the cap. By trying to increase the brightness of the light source and the exposure time of the camera, the brightness of the side wall in the image can be enhanced, but the brightness of the central area has been overexposed. Thus, the image cannot be directly used for subsequent algorithmic processing. Therefore, this paper uses experimental means to adjust the height of the light source by observing the change in the background of the image to be detected. And when the working distance of the light source is about $40mm$, the interior of the cap forms a more uniform illumination, so that the

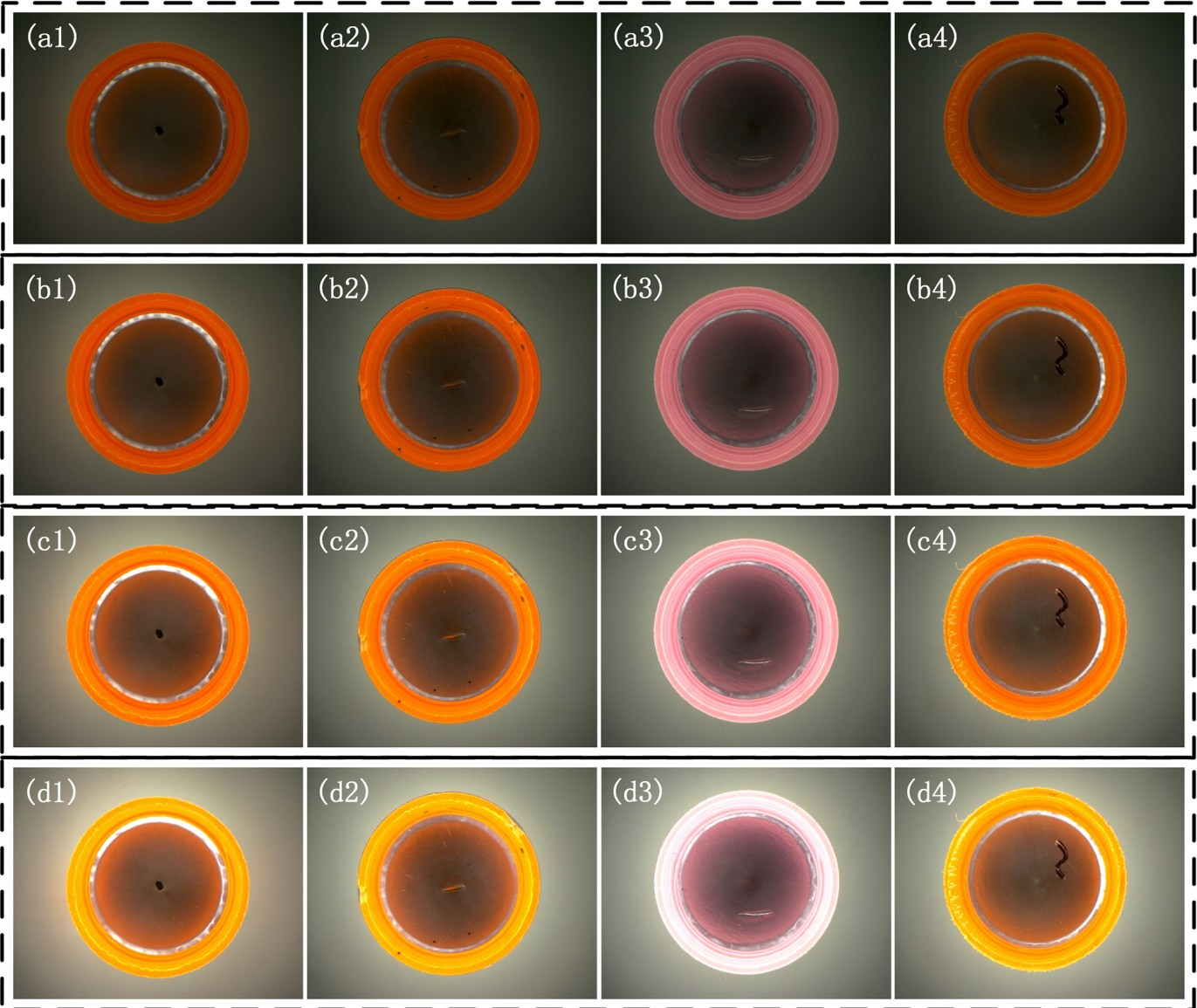

**Fig 13. Comparison of different light intensities in the LASA.** (a) the light source controller is set to 40;(b) the light source controller is set to 70; (c)the light source controller is set to 140; (d)the light source controller is set to 200.

side wall region of the cap has uniform bright-field characteristics, and the bottom inner circle region has uniform dark-field characteristics.

In addition to analyzing the imaging outcomes at varying light source heights, this study also assessed the impact of different light intensities within the LALDA system, as depicted in Fig 13. When the luminance of the light source dips below 70 units (as indicated by the light source controller), the overall appearance of the bottle cap becomes excessively shadowy, leading to challenges in discerning the black spots and scratches on the bottom of the bottle cap. Evidently, the delineation of wrinkles along the gasket's periphery and the demarcation of the dark field boundary at the base become scant, potentially resulting in segmentation inaccuracies in subsequent stages. Conversely, when the light intensity surpasses 140 units, the inner

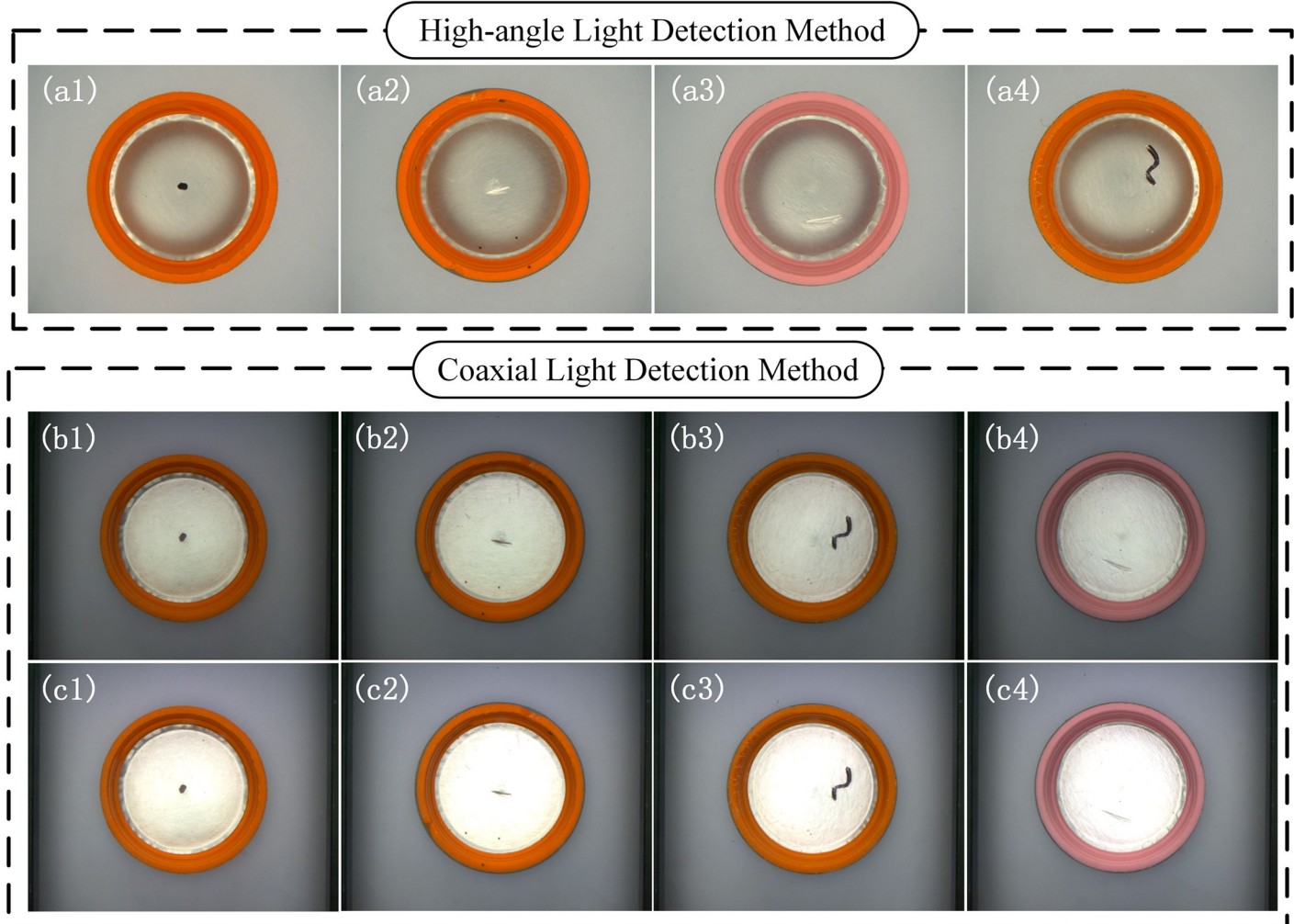

**Fig 14. Comparison of traditional detection systems.** (a) the High-angle Light Detection Method;(b)the Coaxial lighting mode with the light source controller set to 90 degrees;(c) the High-angle Light Detection Method;(b)the Coaxial lighting mode with the light source controller set to 140 degrees.

wall of the bottle cap becomes overexposed, obscuring the defect information within the corresponding area, thereby jeopardizing the detection area during subsequent segmentation processes. Moreover, the high intensity of the light source also affects the dark field effect at the bottom of the bottle cap. Optimal results are achieved when the light intensity ranges between 70 and 140 units, evenly illuminating both the sidewall and bottom regions of the bottle cap. This lighting configuration engenders bright field illumination on the inner sidewall, accentuating the presence of black spot defects. Simultaneously, the base of the bottle cap benefits from dark field illumination, establishing clear delineations of black spot defects, while scratches and wrinkles manifest in a contrasting white hue. Hence, the proposed LALDA system attains superior lighting effects within the optimal light intensity range of 70 to 140 units, showcasing versatile applicability across various scenarios.

In addition, this paper also compared the high-angle light and coaxial light in traditional detection methods [17,18]. In the high-angle light mode, a light source with an incident angle of 60˚ was used, and the results as shown in the Fig 14. Under this light source angle, the

sidewall of the bottle cap can be uniformly illuminated and the defect features are clear, but the illumination at the bottom of the bottle cap is uneven, and partial overexposure occurs, leading to some scratch features being blurred. In the experiment with coaxial light source, the bottom of the bottle cap presents a bright field effect, and the black spot defects are clear, but some scratch defects are not obvious, and the small fluctuation range of the light source intensity may result in different degrees of overexposure for gaskets with different smoothness under the same light source intensity. Lowering the brightness of the light source will result in the inner wall of the bottle cap being too dark, making it difficult to detect black spot defects. Therefore, the LALDA system constructed in this study can provide good illumination for both the inner wall and bottom of the bottle cap.

In summary, the LALDA system proposed in this paper achieves uniform illumination of both the inner wall and bottom of the bottle cap, resulting in excellent imaging effects and a wide range of adjustment in light source intensity compared to traditional detection methods. After obtaining different imaging features through the aforementioned system, further the multi-channel segmentation algorithm (MCS) is used to segment the sub-regions of the image, as well as to homogenize the image and enhance the contrast, and to complete the extraction of defects. The process for different defects detection is shown below.

**(1) Detection of dark point and color mixing**

Firstly, the original image is transformed into HSV color space, and the images of three channels $I_h$, $I_s$, $I_v$ are obtained respectively. For the S-channel, the whole cap region to be detected has a clear contrast with the surrounding background, so the region of the whole cap ($Region_{cap}$) is separated from the image $I_s$. Then, since the dark-field characteristics of the inner circular region at the bottom of the cap are preserved in the V-channel and have a significant grayscale difference from the side wall region, it is possible to simply separate the inner circular region $Region_{circle}$ from the whole cap region $Region_{cap}$ according to the binarization operation. Further, the subtraction of the above two regions can yield the side wall subregion $Region_{side}$. In this paper, in order to avoid the interference of the transition area between the two sub-regions of the cap, and considering that the transition area is mainly the edge of the bottom gasket, the radius of the $Region_{circle}$ is reduced by 30 pixels, and the radius of the circular $Region_{side}$ is reduced by 30 pixels. In addition, for the V-channel, which retains rich detail information as well as high contrast defects, the defects have been detected in the image of V-channel.

In this case, a uniform image $I_{v\_illu}$ is obtained based on operations such as mean filtering and illumination correction ($K_{illu}$ = 0.5), and a high contrast image $I_{v\_em}$ is further obtained by image enhancement ($K_{em}$ = 5). The sub-regions $Region_{side}$ and $Region_{circle}$ mentioned above, are then combined to obtain the image $I_{side}$ and $I_{circle}$ separately. For the dark points in the image $I_{side}$, the segmentation of defects can be achieved directly by adaptive binarization operation (OTSU) to obtain the image $I_{side\_dark}$. For the dark points present in the image $I_{circle}$, this paper subtracts the image $I_{circle}$ to be detected from the standard inner circle image $I_{circle\_std}$, and then combines the binarization to complete the fast defect segmentation to get the image $I_{circle\_dark}$. The image $I_{circle\_std}$ is also obtained by the standard image (image without defects) after the above-mentioned filtering, homogenization, contrast enhancement and region segmentation operations. In addition, since the H-channel retains the hue information of the image, and the color mixing is mainly manifested in hue variations within the $Region_{side}$. In this paper, by calculating the average grayscale $I_{side\_color}$ at the non-zero position in the region $Region_{side}$, the existence of color mixing defect can be quickly determined. The values of $I_{side\_color}$ for the two caps shown in the figure are 7.7 and 168.2 respectively, with significant differences. According to multiple sets of experiments the difference threshold between the two can

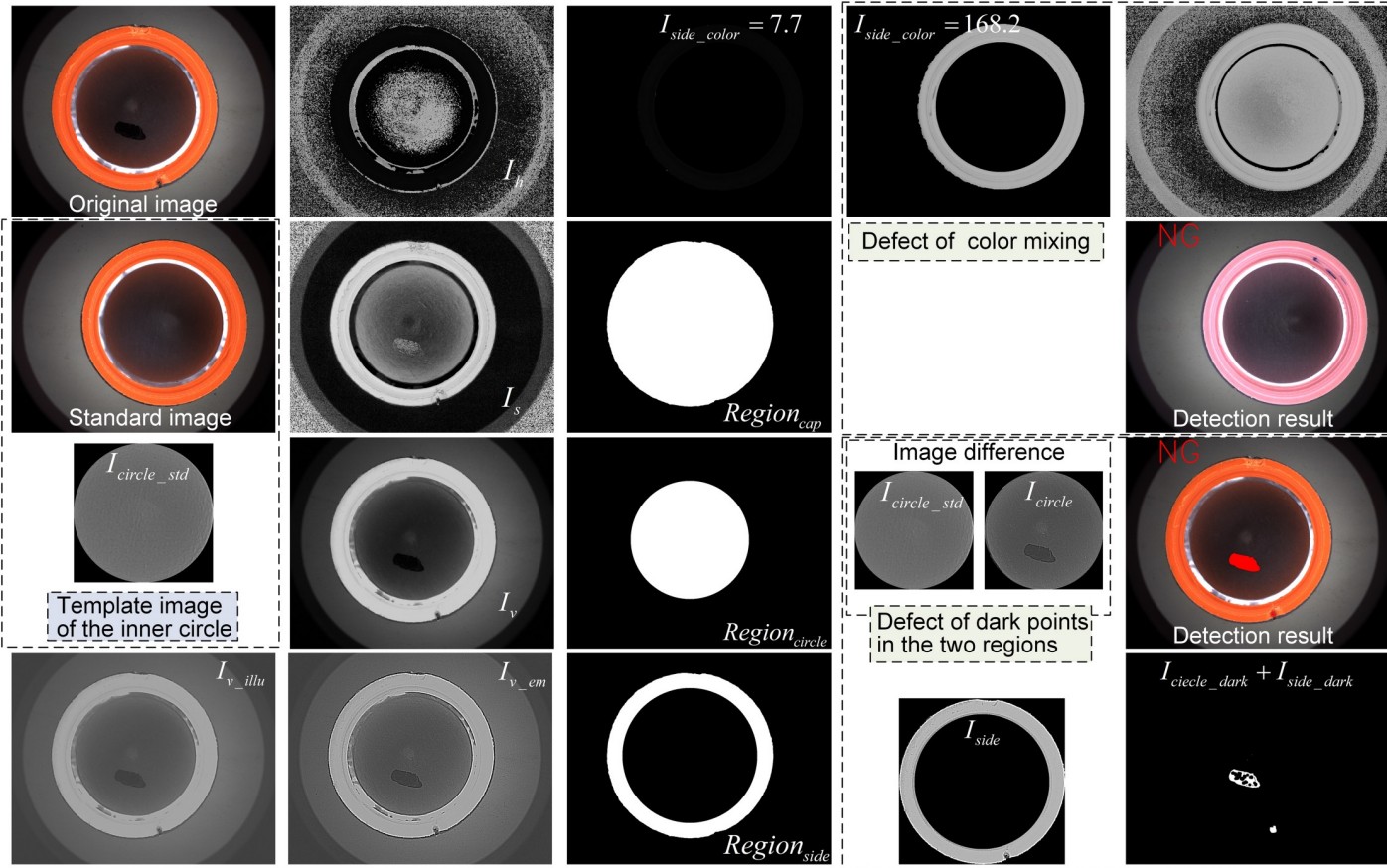

**Fig 15. Detection of dark points and color mixing in the interior of cap.**

be set to 100 to determine whether there is a color mixing defect. Also, it can be manually modified to meet different testing standards. The experimental procedure mentioned above is shown in Fig 15.

**(2) Detection of scratch in the inner circle region**

For the detection of defects in the inner circle region of the bottle cap, the extraction and identification of defects can be achieved mainly based on the subtraction of the detection region from the template image, combined with adaptive binarization and Blob analysis. From the experimental procedure in Fig 16, it can be seen that the inspection system creates a dark field imaging in the inner circle region of the cap, i.e., the background is black and the defect is white. In the enhanced image $I_{circle}$, since the outer contours of the scratch have abrupt changes in grayscale, the grayscale values around the scratch are reduced and lower than the uniform background grayscale values by the contrast enhancement operation. Thus, the outer contour image $I_{circle\_dif}$ of the scratch can be directly obtained by template subtraction, and further by adaptive binarization, the segmentation $I_{circle\_scratch}$ of the defect can be achieved quickly.

**(3) Detection of gasket missing in the inner circle region**

For the defect detection of gasket missing in the inner circle region, a reverse operation using Eq (3) based on the analysis of Section 3 is able to enhance the lower gray values in the

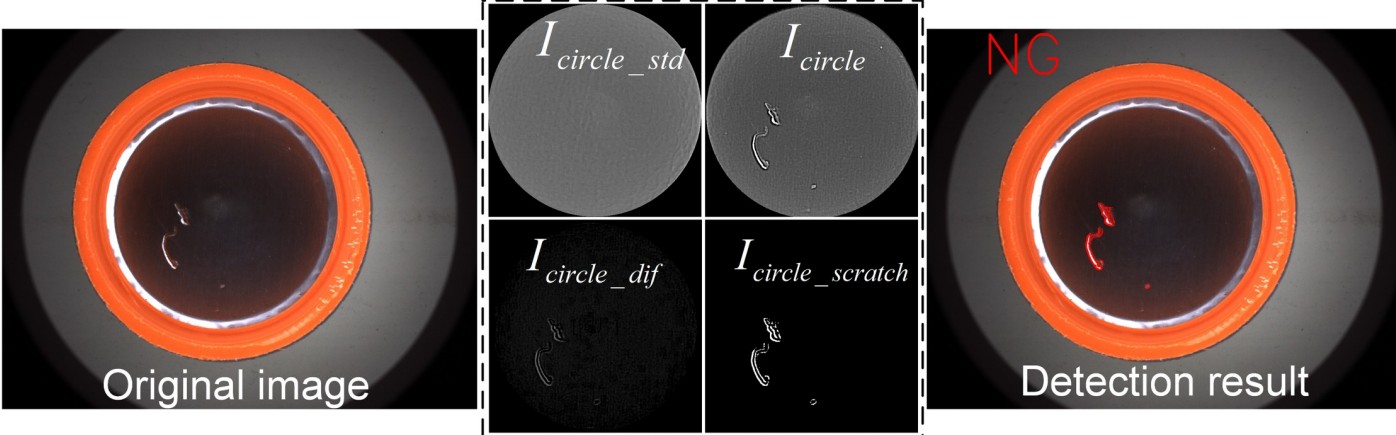

**Fig 16. Detection of scratch in the inner circle region.**

background to brighter gray values ($K_{illu}$ = 3). For the standard image without gasket missing, the inner circle area is darker, and the gray value of the inner circle area is greatly enhanced by the reverse operation. For the image to be detected with gasket missing, the area below the gasket is not smooth, thus generating scattered light and corrupting the dark field imaging, which in turn enhances the background brightness of the inner circle region to some extent. For the standard image without gasket missing, the inner circle area is darker, and the gray value of the inner circle area is greatly enhanced by the reverse operation. For the image to be detected with gasket missing, the area below the gasket is not smooth, thus generating scattered light and corrupting the dark field imaging, which in turn enhances the background brightness of the inner circle region to some extent. On the contrary, its background grayscale is not greatly enhanced by the subsequent operations. By direct binarization (the threshold value chosen *Thresh* = 250 in this paper), the two results of $I_{circle2\_bw}$ can produce large difference. Note that in order to extract the size of the overexposed area, the threshold is set to a higher value of 250. Further calculate the area ($sum_{circle}$) of the bright background present inside the region as $6.0 \times 10^7$ and $1.4 \times 10^7$. Finally, by numerical comparison, $3 \times 10^7$ (by experimental statistics) is used as the threshold for determining whether there is a gasket missing defect. The experimental procedure mentioned above is shown in Fig 17.

**(4) Detection of gasket pleat in the inner circle region**

Due to the defect of gasket pleat, the surface shape of gasket changes, the condition of dark-field lighting is damaged, and bright spot is formed in the black background image. Then, after the homogenization and image enhancement operations, if there is a gasket pleat defect in the inner circle region, there will be a bright region so that the region $I_{circle\_pleat}$ can be segmented directly by binarization (the threshold value *Thresh* = 150 is chosen in this paper). The experimental procedure mentioned above is shown in Fig 18.

Further, the size of the defect is calculated based on the number of bright pixels after binarization. In addition, in actual use, for minor gasket pleat, it can be deemed as conforming. Therefore, based on experimental statistics, $count_{circle} > 5000$ is used as the basis for determining the presence or absence of gasket pleat defect.

**(5) Detection of different defects in bottle caps of different sizes and colors**

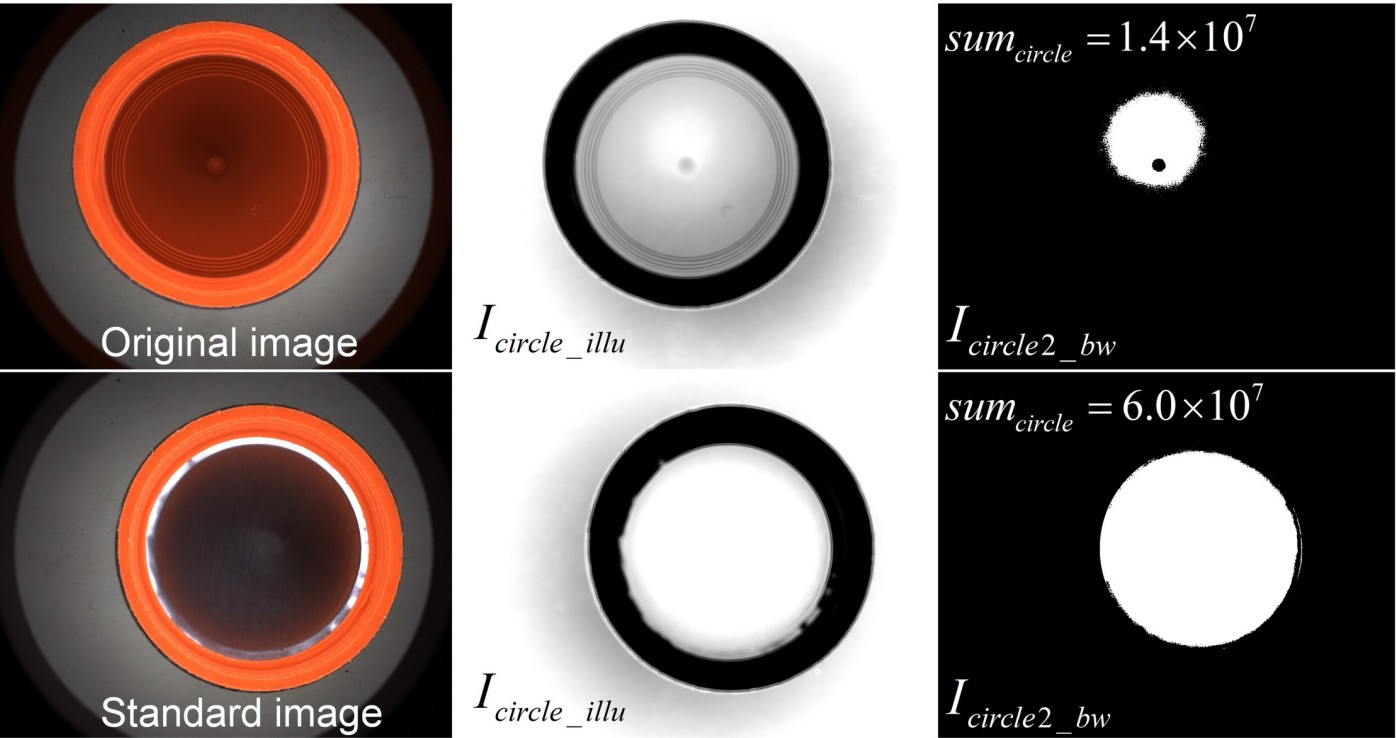

**Fig 17. Detection of gasket missing in the inner circle region.**

In this paper, different kinds of bottle caps are detected in different application scenarios, and the detection results are shown in Fig 19.

**(6) Comparison and Discussion**

In addition, in order to compare the performance with deep learning-based detection methods, this paper collects a total of 300 sample images representing five defect types: namely, blending, dark spots, scratches, missing gaskets, and gasket wrinkles. Among these, 260 images with defects, while 40 are normal images. On the above dataset, this paper compares the multi-channel segmentation (MCS) algorithm designed in this paper with two deep learning network

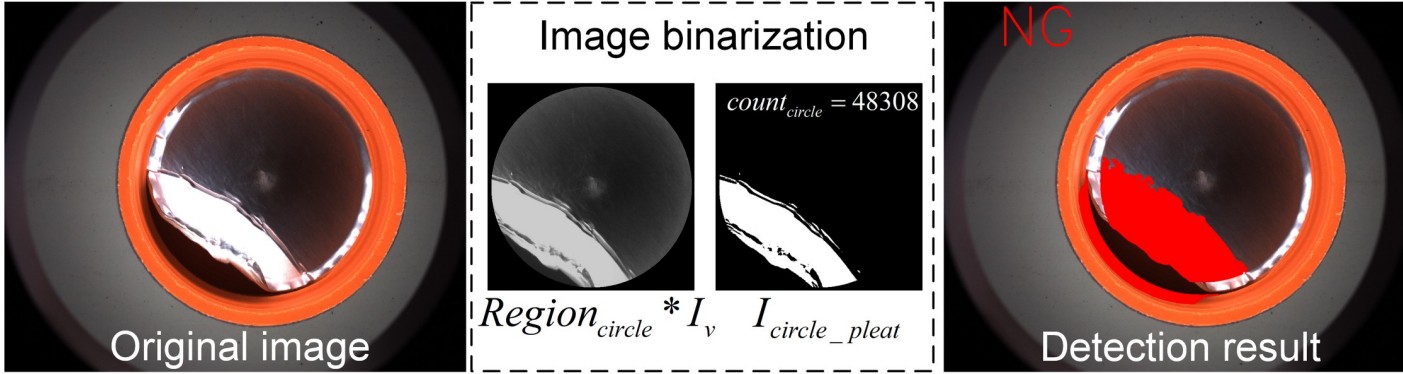

**Fig 18. Detection of gasket pleat in the inner circle region.**

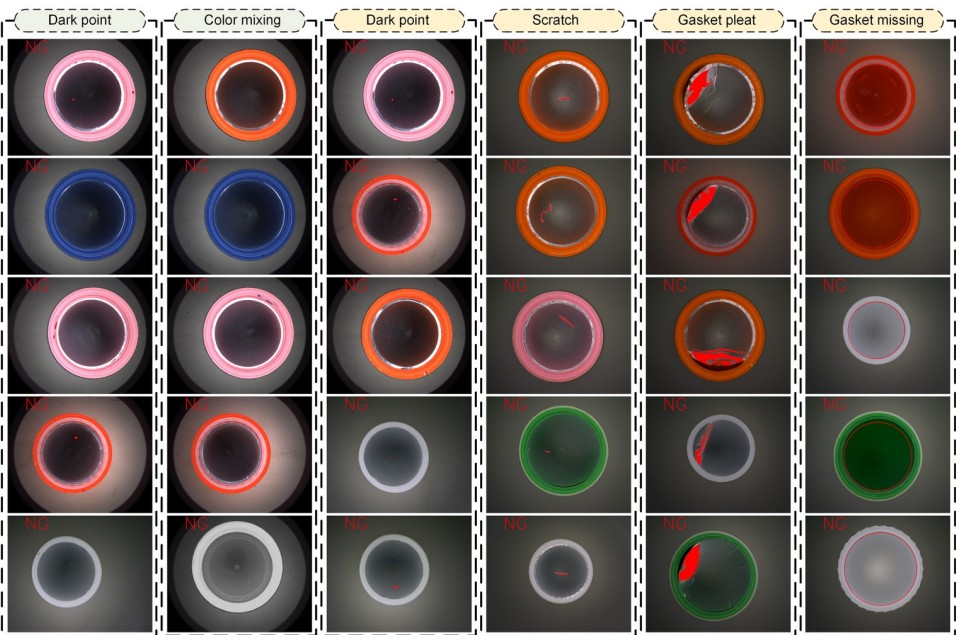

**Fig 19. Detection of different defects in bottle caps of different sizes and colors.**

model based defect detection algorithms, CSEPNet [33] and PoolNet [34]. For both deep learning methods, this paper implemented experiments with the PyTorch framework on a computer equipped with an NVIDIA 4090 GPU. During training, the parameters were updated using the Adam optimizer. In addition, the initial learning rate, total epochs, and batch size were set to $1 \times 10^{-4}$, 100, and 4, respectively. the learning rate was reduced by a factor of 2 for every 20 epochs and multiples of 20 epochs during training. The other preprocessing is consistent with the method in this paper. In the evaluation of deep learning results, Pixel Accuracy (PA), Mean Accuracy (MA) and Mean Intersection over Union (mIoU) are generally selected as the evaluation metrics[35]. PA represents the proportion of correctly segmented pixels to the total number of pixels in the image. It provides a measure of pixel-level classification accuracy for the entire image. MA represents the average of the pixel accuracies for all classes of objects. The mIoU represents the degree of overlap between the segmentation result and the ground truth of the original image. Due to the fact that PA and MA are mainly used to evaluate pixel-level classification accuracy, they are not sensitive enough to boundary information and the accuracy of recognizing different categories, which makes them easily influenced by the number of different defect samples in bottle cap detection. In practical bottle cap detection, greater emphasis is placed on the accuracy of identifying the presence of defects rather than accurately recognizing the types of defects, thereby ensuring the yield of qualified products in production. MIoU comprehensively considers the accuracy of segmentation results and boundary information, enabling better detection of caps with defects. Therefore, considering the inconsistency in the number of defect samples in practical production and to ensure the accuracy of defect presence judgment in real production, this paper selects mIoU as the evaluation metric. It compares the images with an Intersection over Union (IoU) greater than 0.5 with the correct samples to calculate the accuracy of the samples. The final accuracies for the two deep learning methods are 68.4% and 73.7%, respectively, as shown in Table 1. The experimental results show that under the same data sample, the accuracy of the deep learning

**Table 1. Comparison of detection accuracy of different detection methods.**

| Method | Average Accuracy |
|---|---|
| MCS in this paper | 97.3% |
| CSEPNet [33] | 68.4% |
| PoolNet [34] | 73.7% |

method is significantly lower than that of our proposed multi-channel segmentation (MCS) detection algorithm. The detection results of some misidentifications in deep learning are shown in Fig 20.

The machine vision system proposed, through the statistical analysis of actual accuracy rates, achieves an average detection accuracy of about 97.3% for various defects. Due to the challenges of detecting inconspicuous dark point and scratches, as well as cases where colors are extremely similar, there are instances of false positives and false negatives. However, the overall accuracy rate is superior to the accuracy requirements of actual scenarios, which is 95%. At the same time, through actual testing, the average time taken for detecting defects inside the bottle cap is about 0.13s, which is better than the efficiency (400 pcs/min) required in practical application scenarios. In addition, the detection resolution of this paper is about

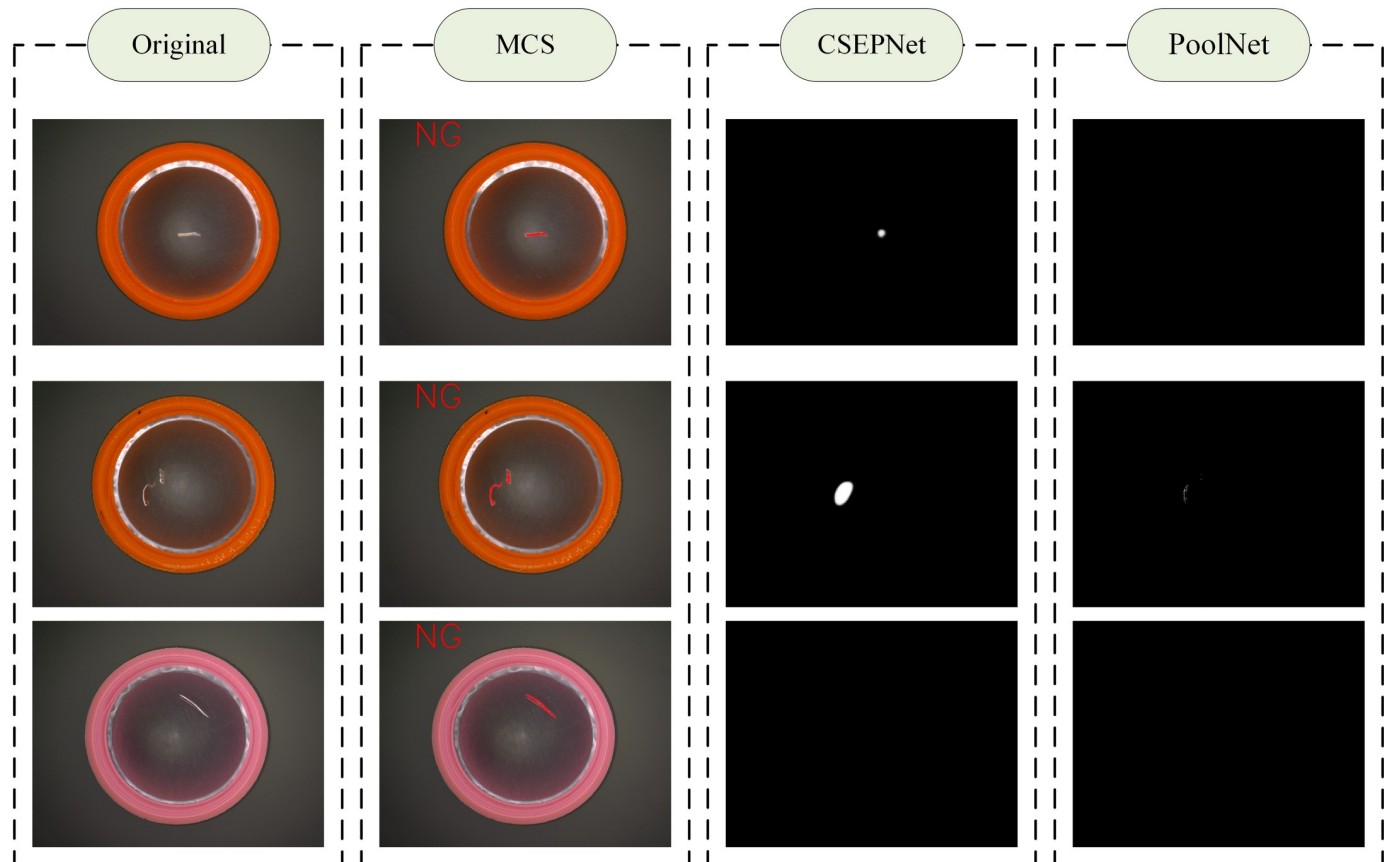

**Fig 20. Comparison of results: Small black spots and scratches, as well as the shape, direction, and curvature of the scratches, have a certain randomness, leading to missed detections.**

0.5*mm*, which is better than the required detection resolution (1*mm*). Besides, in this paper, some simple judgment conditions such as area thresholds are used in identifying defects, which can be quickly modified to meet the requirements of different degrees of detection in different inspection scenarios. For example, if different manufacturers have different requirements on the degree of gasket pleat when detecting gasket pleats, the algorithm can be deployed directly by modifying the corresponding determination conditions, thus improving the compatibility of the algorithm. Therefore, the vision inspection system based on low angle and large divergence angle designed in this paper can meet the actual inspection requirements, and can be quickly adapted to the new product, as well as quickly matched to the production line.

In addition, the application scenario in this paper focuses on the detection of defects in the more common round bottle caps of different sizes and colors. According to its specific shape, the ring illumination with low angle and large divergence angle is used to be able to form a uniform imaging. However, for other shapes of caps, specific lighting methods for special shapes are carried out in later application studies. In addition, for the current stage of research, the detection system designed in this paper (including the hardware system for detection and the detection algorithm) has been able to meet the actual scenario application, in some calculations need to calculate some parameters based on some data, and need to make some detection templates, etc., which brings some inconvenience. The methods such as deep learning can further improve the generality of the whole system, but at this stage, due to the high production yield in the actual scenario, there are fewer samples with defects, which will result in lower detection accuracy shown in Table 1. Therefore, the design of the defect recognition algorithm in the system is currently completed for the characteristics of the defect, based on the classical algorithm and according to the obvious feature differences. With the increase in data on products with defects in the subsequent use of the equipment (LALDA proposed in this paper), the versatility of the detection equipment will be further enhanced in the subsequent research based on deep learning methods.

## 5. Conclusion

In this paper, the defect detection inside the cap of a pharmaceutical bottle is the object of study. Due to its own shape characteristics, it is easy to cause light blockage in the illumination light path. Existing detection techniques cannot meet uniform illumination and the algorithm part cannot achieve a more generalized detection. This paper proposes a low-angle and large dispersion angle vision inspection system (LALDA), using the large dispersion angle of LED, combined with low-angle illumination, to achieve uniform bright-field illumination of the side wall of the bottle cap and uniform dark-field illumination of the inner circle region at the bottom. Compared to existing methods, it can avoid the problem of the illumination blockage of the bottom inner circle by the side of the bottle cap with high height, and solves the problem of low brightness and low contrast of the side wall of the bottle cap in the traditional detection schemes. Further, based on the distinctive distinguishing features in the uniform image, a multi-channel segmentation algorithm (MCS) is designed for the imaging characteristics of the bottle cap region. The adaptive segmentation of the sub-regions to be processed is accomplished by mutual calculation between different channels of HSV. Then, the background in different sub-regions is further homogenized and the contrast of defects is further enhanced by homogenization and enhancement operations. Based on the above methods and image pre-processing, the subsequent extraction and identification of defects can be quickly achieved by simple gray-scale segmentation and Blob analysis for different types of defects existing in

different sub-regions. Finally, the experiments show that the detection method proposed in this paper can meet the capacity of the actual production line as well as the detection requirements.

## Supporting information

**S1 File.**
(RAR)

## Author Contributions

**Conceptualization:** Bowen Chen, Chen Li, Pi Yuan, Yongjing Yin.

**Data curation:** Bowen Chen, Chen Li, Pi Yuan, Yongjing Yin.

**Formal analysis:** Chen Li.

**Investigation:** Chen Li, Pi Yuan, Yujie Yan.

**Methodology:** Chen Li, Pi Yuan, Yujie Yan.

**Supervision:** Yujie Yan, Yongjing Yin.

**Validation:** Bowen Chen, Chen Li, Pi Yuan.

**Writing – original draft:** Chen Li, Pi Yuan.

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
