## [Decision Letter · Decision Letter 0]

26 Mar 2024

PONE-D-24-01223Research on defect detection of bottle cap interior based on low-angle and large divergence angle vision systemPLOS ONE

Dear Dr. Li,

Thank you for submitting your manuscript to PLOS ONE. After careful consideration, we feel that it has merit but does not fully meet PLOS ONE’s publication criteria as it currently stands. Therefore, we invite you to submit a revised version of the manuscript that addresses the points raised during the review process.

We look forward to receiving your revised manuscript.

Kind regards,

Alessandro Bruno, Ph.D.

Academic Editor

PLOS ONE

Journal Requirements:

5. Please upload a copy of Supporting Information Figure/Table/etc. "S1 File (ZIP)" which you refer to in your text on page 26.

**Additional Editor Comments:**

Dear Authors,

Your manuscript shows scientific soundness with some exhaustive experimental campaigns.

However, I recommend your paper for a minor revision round as for you to address some minor issues.

You're invited to deal with all the reviewer's suggestions.

Please, run a dry English language check before getting through the next step.

Sincerely,

A.B.

Reviewers' comments:

Reviewer's Responses to Questions

**Comments to the Author**

1. Is the manuscript technically sound, and do the data support the conclusions?

Reviewer #1: Yes

2. Has the statistical analysis been performed appropriately and rigorously? 

Reviewer #1: Yes

3. Have the authors made all data underlying the findings in their manuscript fully available?

Reviewer #1: Yes

4. Is the manuscript presented in an intelligible fashion and written in standard English?

Reviewer #1: Yes

5. Review Comments to the Author

Reviewer #1: Minor Revision

(1)It is important to address whether the luminous intensity of the light source has any impact on the test results, and if so, provide a thorough explanation in the manuscript.

(2)The authors may add more state-of-art computer vision & robot articles in engineering application for the integrity of the manuscript (3D vision technologies for a self-developed structural external crack damage recognition robot; Automation in Construction. Enhanced Precision in Dam Crack Width Measurement: Leveraging Advanced Lightweight Network Identification for Pixel-Level Accuracy; International Journal of Intelligent Systems.).

(3)In the experimental section, the average detection correct rate is reported as 97.3% and compared with the average accuracy results of other deep learning networks in Table 1. However, it's crucial to acknowledge that the average accuracy of the deep learning network is just one type of network evaluation index and may not be directly comparable to the average detection correct rate. This distinction should be carefully considered and appropriately addressed.

(4)The paper's detection method and algorithms represent a departure from traditional detection schemes. Therefore, it is advisable to expand the comparison with traditional detection schemes to demonstrate the improvement in the correct detection rate.

(5)Provide a clear definition of "defective caps" to ensure readers have a complete understanding of the terminology used in the study.

6. PLOS authors have the option to publish the peer review history of their article (what does this mean?). If published, this will include your full peer review and any attached files.

Reviewer #1: No

---

## [Author Response · Author response to Decision Letter 0]

30 Apr 2024

Reviewer#1, Concern # 1: It is important to address whether the luminous intensity of the light source has any impact on the test results, and if so, provide a thorough explanation in the manuscript.

Author response: Thank you for your comments. Comparative experiments with different light source intensities have been added to confirm that the LALDA system works equally well over a wide range of light intensities.

Author action: We have supplemented the relevant experiments and updated the manuscript.

Reviewer#1, Concern # 2: The authors may add more state-of-art computer vision & robot articles in engineering application for the integrity of the manuscript (3D vision technologies for a self-developed structural external crack damage recognition robot; Automation in Construction. Enhanced Precision in Dam Crack Width Measurement: Leveraging Advanced Lightweight Network Identification for Pixel-Level Accuracy; International Journal of Intelligent Systems.).

Author response: Thank you for your comments. We added relevant literature studies.

Author action: We have updated the manuscript.

Reviewer#1, Concern # 3: In the experimental section, the average detection correct rate is reported as 97.3% and compared with the average accuracy results of other deep learning networks in Table 1. However, it's crucial to acknowledge that the average accuracy of the deep learning network is just one type of network evaluation index and may not be directly comparable to the average detection correct rate. This distinction should be carefully considered and appropriately addressed.

Author response: Thank you for your comments. As you mentioned, our study does not adequately take into account the comparison of other evaluation metrics in deep learning, nor does it clearly define the average accuracy. In view of this, we have supplemented the study as follows, and described accordingly in the manuscript: pixel accuracy (PA) and average accuracy (MA) are mainly used to evaluate classification accuracy, but in actual cap production, more attention is paid to the accuracy of defect detection, that is, whether defects can be accurately identified to meet the manufacturer's 95% accuracy requirement (at least 95 out of 100 defective caps can be correctly identified). Therefore, we chose IoU (Intersection over Union) as the evaluation index in the paper to measure accuracy more accurately.

Author action: We have updated the manuscript.

Reviewer, Concern # 4: The paper's detection method and algorithms represent a departure from traditional detection schemes. Therefore, it is advisable to expand the comparison with traditional detection schemes to demonstrate the improvement in the correct detection rate.

Author response: Thank you for your comments. We supplemented the comparison experiments of high-angle light sources and coaxial light sources in the traditional method, and analyzed the existing problems.

Author action: We have supplemented the relevant experiments and updated the manuscript.

Reviewer, Concern # 5: Provide a clear definition of "defective caps" to ensure readers have a complete understanding of the terminology used in the study.

Author response: Thank you for your comments. We have made appropriate changes and explanations to the relevant descriptions.

Author action: We have updated the manuscript.

---

## [Editor Report · Decision Letter 1]

1 May 2024

Research on defect detection of bottle cap interior based on low-angle and large divergence angle vision system

PONE-D-24-01223R1

Dear Dr. Li,

We’re pleased to inform you that your manuscript has been judged scientifically suitable for publication and will be formally accepted for publication once it meets all outstanding technical requirements.

Kind regards,

Alessandro Bruno, Ph.D.

Academic Editor

PLOS ONE

Additional Editor Comments (optional):

Dear Authors,

I appreciate your efforts in dealing with the reviewer's remarks and suggestions.

I believe your paper is now ready for acceptance although there are still some typos.

Hereby, I recommend your manuscript for acceptance.

Kinde regards
---

## [Editor Report · Acceptance letter]

8 May 2024

PONE-D-24-01223R1 

PLOS ONE

Dear Dr. Li, 

I'm pleased to inform you that your manuscript has been deemed suitable for publication in PLOS ONE. Congratulations! Your manuscript is now being handed over to our production team.

Kind regards, 

on behalf of

Senior Assistant Professor Alessandro Bruno 

Academic Editor

PLOS ONE